# Expression of *Talaromyces marneffei*
*acuM* and *acuK* Genes in Gluconeogenic Substrates and Various Iron Concentrations

**DOI:** 10.3390/jof6030102

**Published:** 2020-07-08

**Authors:** Monsicha Pongpom, Artid Amsri, Panwarit Sukantamala, Phimchat Suwannaphong, Juthatip Jeenkeawpieam

**Affiliations:** Department of Microbiology, Faculty of Medicine, Chiang Mai University, Chiang Mai 50200, Thailand; artid_a@cmu.ac.th (A.A.); jedsada_suk@cmu.ac.th (P.S.); phimchat.s@cmu.ac.th (P.S.); Juthatip_jee@cmu.ac.th (J.J.)

**Keywords:** gluconeogenesis, iron

## Abstract

*Talaromyces marneffei* is an opportunistic, dimorphic fungal pathogen that causes a disseminated infection in people with a weakened immunological status. The ability of this fungus to acquire nutrients inside the harsh environment of the macrophage phagosome is presumed to contribute to its pathogenicity. The transcription factors AcuM and AcuK are known to regulate gluconeogenesis and iron acquisition in *Aspergillus fumigatus*. This study demonstrated that they are also involved in both of these processes in the dimorphic fungus *T. marneffei*. Expression of *acuM* and *acuK* genes was determined by real time-polymerase chain reaction (RT-PCR) on the cells grown in media containing gluconeogenic substrates and various iron concentrations. We found that the *acuM* and *acuK* transcript levels were sequentially reduced when growing the fungus in increasing amounts of iron. The *acuM* transcript was upregulated in the gluconeogenic condition, while the *acuK* transcript showed upregulation only in the acetate medium in the yeast phase. These results suggest the involvement of *acuM* and *acuK* in gluconeogenesis and iron homeostasis in *T. marneffei*.

## 1. Introduction 

*Talaromyces marneffei* is an opportunistic fungus that causes a disseminated infection in immunocompromised patients in Southeast Asian countries and people who travel into this area of endemicity [1,2]. Infection due to *T. marneffei* is presumed to begin with multiple factors affecting the growth of the fungus inside the host immune cells. After the inhaled conidia reach the alveoli, they are phagocytosed by the alveolar macrophages. The conidia can resist the weakened killing mechanism and convert the growth pattern into a yeast phase. Then, *T. marneffei* yeast cells can replicate and survive within the alveolar macrophages. As a facultative, intracellular pathogen, a *T. marneffei* infection requires the ability to obtain the nutrients necessary for growth and replication under the nutrient-deprived conditions inside the host cells for successful establishment. 

Utilization of gluconeogenic carbon sources is one of the ways fungi obtain energy from the environment. This ability allows the fungal pathogen to survive and replicate inside the host. Gluconeogenesis is one of the mechanisms by which intracellular fungal pathogens can acquire carbon. Gluconeogenesis produces glucose from certain noncarbohydrate carbon substrates, and then the glucose is used in cellular metabolism [3,4,5]. Disruption of the genes involved in gluconeogenesis attenuates fungal virulence [6,7,8]. In *Aspergillus fumigatus*, disruption of the regulators that control gluconeogenesis also affects the virulence in the murine model [9,10]. 

The survival of the fungal pathogens within the host also depends on iron micronutrients, which play a role in most metabolic processes, mainly functioning as enzyme cofactors. The host restricts iron via a process called nutritional immunity [11]. Thus, a competition for iron usually occurs between the host and pathogens. Fungi have developed several mechanisms to obtain sufficient iron from the host. For instance, *A. fumigatus* acquires iron from the host via reductive iron assimilation (RIA) and siderophore biosynthetic pathways. Siderophore-assisted iron uptake has been shown to be essential for pathogenicity [12,13]. RIA and siderophore-mediated iron uptake also play a role in iron acquisition in the yeast pathogenic form of *T. marneffei* [14]. 

Fungal pathogens have developed sophisticated mechanisms to control the iron homeostasis [15,16]. Iron assimilation mechanisms have been well studied in *A. fumigatus.* The most important controlling system components are transcription factors SreA and HapX, which regulate iron consumption and acquisition pathways directly [13,17,18,19]. Recently, transcription factors AcuM and AcuK have also been reported to control siderophore production by inhibition of SreA, resulting in enhanced uptake of iron via siderophores [9,10].

AcuM and AcuK are homologous Zn(2)Cys(6) transcription factors that were previously known as the regulators for gluconeogenesis in *A. nidulans*. AcuM and AcuK have been demonstrated to work as a heterodimer [20]. Interestingly, they have also been demonstrated to work divergently and to control both gluconeogenesis and iron metabolism in *A. fumigatus* [10]. Despite the highly conservative nature of AcuM and AcuK and their closely related evolutions in *A. nidulans* and *A. fumigatus*, these transcription factors play diverse and unexpectedly distinct roles in these two *Aspergilli*. This led to the question of how these transcription factors function in the dimorphic fungus *T. marneffei*. 

Bioinformatics analysis showed that *T. marneffei*’s genome contains *acuM* and *acuK* genes. Thus far, the role of *acuM* and *acuK* in *T. marneffei* has been unknown. This study focused on the role of *acuM* and *acuK* in gluconeogenesis and iron assimilation. We charted the growth characteristics of *T. marneffei* and the expression patterns of *acuM* and *acuK* in the presence of gluconeogenic substrates and various iron conditions. Understanding the dynamics and effects of nutrient assimilation could provide valuable insight into the process by which *T. marneffei* infects humans.

## 2. Materials and Methods

### 2.1. Fungal Strain and Culture Conditions

*Talaromyces marneffei* ATCC200051 (clinically isolated strain, Chiang Mai, Thailand, 1996) was grown on a malt extract agar at 25 °C. The conidial suspension was prepared by scraping the conidia from the surface of the mold colony with a cotton swab and was then enumerated. 

To examine the growth in gluconeogenic conditions, the following carbon sources were added to a carbon-free medium (containing, per liter, the salt solution 6 g NaNO_3_, 0.52 g KCl, 0.52 g MgSO_4_·7H_2_O, and 1.52 g KH_2_PO_4_; Hutner’s trace element 2.2 g ZnSO_4_·7H_2_O, 1.1 g H_3_BO_3_, 0.5 g MnCl_3_·4H_2_O, 0.5 g FeSO_4_·7H_2_O, 0.16 g CoCl_2_·6H_2_O, 0.16 g CuSO4·5H_2_O, 0.11 g (NH_4_)MO_7_O_24_·4H_2_O, and 5 g ethylenediaminetetraacetic acid (EDTA); and 10 mM ammonium sulfate as a nitrogen source): 50 mM proline, 0.5% ethanol, and 50 mM acetate. A 3-μL suspension containing 10^8^ conidia of *T. marneffei* was dropped onto the agar surface and incubated at either 25 °C or 37 °C for colony observation. 

To determine the growth in various iron concentrations, a concentration of 10^8^ conidia/mL was prepared in a glucose minimal medium without iron. To generate an iron-depleted condition (0 mM iron), 100 μM phenanthroline was added to the medium. To generate media with various iron concentrations, the iron-depleted medium was supplemented with a ferric chloride (FeCl_3_) solution to the final concentrations of 0.01, 0.10, 1.00, and 2.00 mM. The *T. marneffei* conidia (10^8^ conidia/3 μL) was spotted onto the agar surface and incubated at either 25 °C or 37 °C for colony morphology observation. To measure the hyphal mass, the conidia was grown in a 100-mL liquid medium and filtered, baked at 50–60 °C for 1–2 days, and weighed. 

### 2.2. RNA Isolation and cDNA Synthesis

*T. marneffei* was grown at a concentration of 10^8^ conidia/mL in culture media containing gluconeogenic substrates or iron at different concentrations. The cultures were shaken at 150 rpm at either 25 °C for 36 h or 37 °C for 60 h. The fungal cells were harvested by centrifugation and resuspended in a TRIzol reagent (Life Technologies, Carlsbad, CA, USA). The total RNA was extracted by mechanical disruption of the cells in a bead beater (Biospec, Bartlesville, OK, USA). The RNA was treated with DNase to remove the contaminated DNA, and the RNA concentration was determined by spectrophotometry (Nanodrop2000, Thermo Fisher Scientific, Wilmington, DE, USA). To confirm the absence of DNA contamination, a polymerase chain reaction was performed using primers ActF (5′-GGTGATGAGGCACAGTC-3′) and ActR (5′-GAAGCGGTCTGGATCTC-3′). The 600-bp actin-amplified product was not observed in all RNA samples. 

To generate complementary DNA (cDNA), 1 μg of the total RNA was reverse transcribed using a ReverTra Ace qPCR RT Master Mix kit (TOYOBO Inc., Osaka, Japan). Briefly, the RNA sample was denatured at 65 °C for 5 min and then chilled on ice. Then the reverse transcription reaction was performed in a master mix buffer containing oligo-dT primer and reverse transcriptase enzyme at 37 °C for 30 min.

### 2.3. Quantitative Real-Time PCR (qRT-PCR)

An SYBR Green Thunderbird PCR Master Mix (Toyobo Inc., Osaka, Japan) was used for qRT-PCR. Expression of an actin gene was used as an internal control and in calculation of relative expression of *acuM* and *acuK* genes in each sample. Amplifications were performed using Real_AcuM_F primer (5′-GATTCCGGCTTGTTGCTG-3′) and Real_AcuM_R primer (5′-CTTCTGAGAGCCGTCGAATG-3′) for detection of the *acuM* transcript and Real_AcuK_F primer (5′-CCTCCGCCACGGATCATAGTG-3′) and Real_AcuK_R primer (5′-ACACCGTCGTGGCATGCATC-3′) for detection of the *acuK* transcript. PCR programming was 95 °C for 60 s, followed by 40 cycles of 95 °C for 60 s. The PCR products were measured for fluorescence intensity in a 7500 Real-Time PCR System (Applied Biosystems, Foster City, CA, USA). Calculation of the fold change of *acuM* and *acuK* genes normalized to actin and relative to their expression in a glucose condition were performed using the formula 2^−(ΔΔC_t_)^, where ΔΔC_t_ = (C_t_
*_acuM_*
_or *acuK*_ − C_t_ actin)_gluconeogenic substrates_ − (C_t_
*_acu_*_M or *acuK*_ − C_t_ actin)_glucose_ in gluconeogenesis assay. To validate the effect of iron concentrations on the expression of *acuM* and *acuK* on the expression of the actin gene, the relative expressions were calculated using the formula 2^−ΔC_t_^, where ΔC_t_ = C_t_
*_acuM_*
_or *acuK*_ − C_t actin_.

### 2.4. Statistical Analysis

The data were analyzed using analysis of an independent sample paired *t*-test where *p* values of <0.05 were considered significant. All statistical analysis was performed using SPSS v.16.0.

### 2.5. Bioinformatics

Similarity analysis was performed using Blast programs on the NCBI database (Available online: https://blast.ncbi.nlm.nih.gov/Blast.cgi). Selected orthologous sequences were analyzed with Clustal Omega to view the phylogenetic tree of the alignment. 

## 3. Results 

### 3.1. Growth of T. marneffei on Gluconeogenic Substrates

Glucose is the preferential carbon source for *T. marneffei*. However, *T. marneffei* has ability to use various alternative gluconeogenic substrates as its sole carbon source. Growth of *T. marneffei* on media containing different carbon sources is shown in Figure 1 and Figure 2. Macroscopic morphology was observed on the agar medium (Figure 1A). *T. marneffei* produced a green, velvety colony without red-soluble pigment production within 12 days on the glucose-containing medium (control). Growth on the gluconeogenic substrates showed less mycelia mass compared to the glucose medium. On the proline- and ethanol-containing media, the colony diameters and rates of conidiation were moderately reduced compared to the growth on the glucose medium, while the growth and conidial production were dramatically reduced on the acetate medium (Figure 1A, surface). Interestingly, red pigment production was observed in the growth on the gluconeogenic-containing media (Figure 1A, reverse). Hyphal production was observed microscopically (Figure 1B). The hyphal mass in the glucose-containing medium was higher than in the gluconeogenic carbon-containing media. However, the size of the hyphae produced by the fungal cultures of acetate was approximately 2 times larger than that of glucose (5 μM in acetate vs. 2–3 μM in glucose). 

### 3.2. Growth of T. marneffei in Various Iron Concentrations

The effect of different iron concentrations was determined by dry weight and colony diameter measurement when growing the fungus in the liquid and on the solid media, respectively. *T. marneffei* could not grow in iron-depleted agar or broth (0 mM iron, the medium containing phenanthroline). The colony morphology was normal, but the diameters decreased sequentially with higher iron concentrations (Figure 3). Similar results were observed in the dry mass determination assay. In the mold phase, a significant decrease in growth was observed at 1 and 2 mM iron. In the yeast form, a reduction in cell mass with increasing iron concentration up to 2 mM was observed, though the change was not significant (Figure 4). The fungus could not grow in liquid media containing 5 and 10 mM iron. This result indicated that *T. marneffei* had encountered iron toxicity, and this reduced growth was seen more prominently in the broth culture than on the agar media. 

### 3.3. Expression of acuM and acuK in Gluconeogenic and Iron Conditions 

To examine the role of AcuM and AcuK in gluconeogenesis and iron metabolism, the relative expression levels of *acuM* and *acuK* were determined during the fungal growth with different gluconeogenic carbon sources and iron concentrations. In the presence of the gluconeogenic carbon sources, both *acuM* and *acuK* expression showed insignificant upregulation in the mold phase when compared to the glucose (Figure 5A). However, significant differences could be observed in *acuM* expression in the yeast phase (Figure 5B). Upregulation of *acuK* was only observed in the yeast phase in the acetate medium (Figure 5B). These results suggest that *acuM*, and possibly *acuK*, play a conservative role in the control of gluconeogenesis like they do in *A. nidulans* and *A. fumigatus*. 

The increased iron concentrations caused sequential declines in the levels of *acuM* and *acuK* transcripts in both the mold and yeast phases (Figure 6). A significant reduction was observed at 1 mM compared to 0.03 mM iron. This result suggests the possible function of *acuM* and *acuK* in negative control of the genes that are responsible for alleviating iron toxicity. 

## 4. Discussion

Control of gluconeogenesis by *acuM* and *acuK* has been reported in *A. nidulans* [20,21]. However, the genes were recently discovered to have an additional role in control of iron acquisition in *A. fumigatus* [9,10], while they did not provide this function in *A. nidulans*. This divergent function is intriguing since the two orthologs are otherwise highly similar in these two *Aspergilli*. At the present, there has been no attempt from investigators to perform gene replacement in order to see whether the orthologs from *A. fumigatus* and *A. nidulans* could be exchangeable or restorable in their functions. We doubt whether *acuM* and *acuK* function in *T. marneffei* in the same manner as they do in *A. nidulans* or *A. fumigatus*. Our bioinformatics analysis could not answer this question since the products of both genes showed a high level of similarity (more than 60%) to the orthologs in *A. nidulans* and *A. fumigatus* (Appendix A). Investigation into *acuM* and *acuK* expression patterns was thus performed in this study to demonstrate the genes’ involvement in gluconeogenesis and iron metabolism.

*T. marneffei* has an ability to grow on various alternative carbon sources. We observed red pigment production, which indicated the activation of polyketide synthesis when the fungus was growing on gluconeogenic substrates. The red pigment is a mixture of several chemical compounds produced by secondary metabolism [22]. One possible explanation for the red pigment production is that the polyketide biosynthetic pathways were activated via nutrient starvation, as this is one of the general stress responses found in several fungi [23,24]. However, the exact mechanism inducing the synthesis of these polyketides during fungal growth on gluconeogenetic substrates is unknown and needs further investigation.

Investigation into the involvement of *T. marneffei* AcuM and AcuK transcription factors in gluconeogenesis and iron metabolism found their relationship to both pathways to be similar to that found in *A. fumigatus*. A significantly increased level of *acuM* transcript was found in the yeast form during growth on gluconeogenic substrates. The elevation of *acuK*, even though not prominent, also showed a trend of upregulation. This result implies that *acuM*, and possibly *acuK*, in *T. marneffei* has the same conservative role in gluconeogenesis found in *A. nidulans* and *A. fumigatus*. The role of AcuM and AcuK transcription factors in iron metabolism could be to either reduce iron acquisition or alleviate iron toxicity, since they were downregulated in the presence of high iron concentrations. The molecular mechanism in control of iron homeostasis is sophisticated and involves several factors. Further studies should be performed to answer the question of how *acuM* and *acuK* affect iron homeostasis in *T. marneffei*.

In summary, this study provides information on the possible functions of *acuM* and *acuK* in *T. marneffei*. Further mutation experiments will be performed to verify their exact function and confirm these observations. Additionally, experiments should be performed in order to prove whether they assist in intracellular survival ability. If so, they would be ideal drug targets based on two important properties: the virulence factors and fungal-specific proteins. Blocking the transcription factors responsible for fungal adaptation pathways could inhibit infection at an early stage.

## Figures and Tables

**Figure 1 jof-06-00102-f001:**
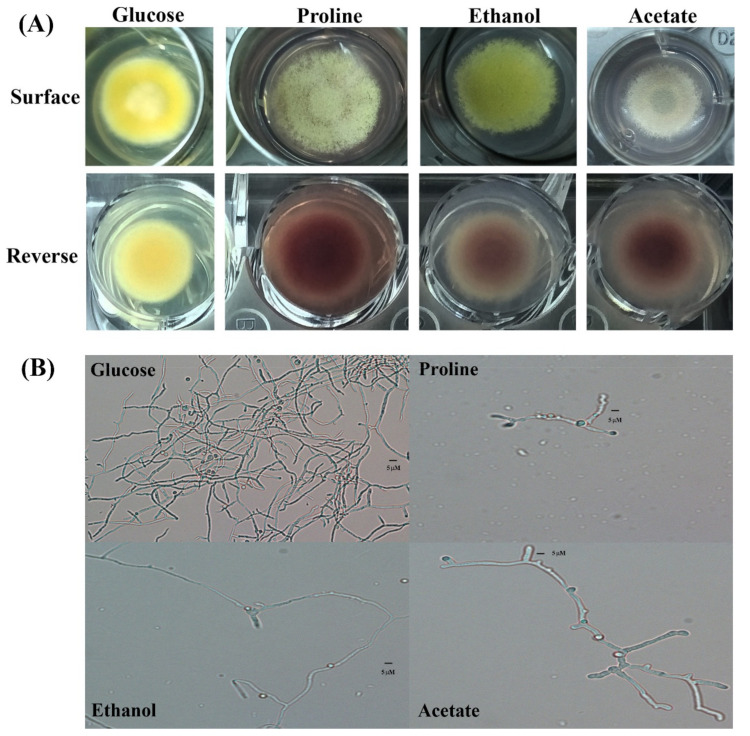
Mold-phase morphology of *Talaromyces marneffei* on gluconeogenic carbon sources. The carbon-free medium was supplemented with 1.0% glucose (preferential carbon source), 50 mM proline, 0.5% ethanol, or 50 mM sodium acetate as the sole carbon source. (**A**) Macroscopic mold colony morphology after incubation at 25 °C for 12 days. Red pigment production is shown on the reverse side of the culture. (**B**) Microscopic morphology (magnification 200).

**Figure 2 jof-06-00102-f002:**
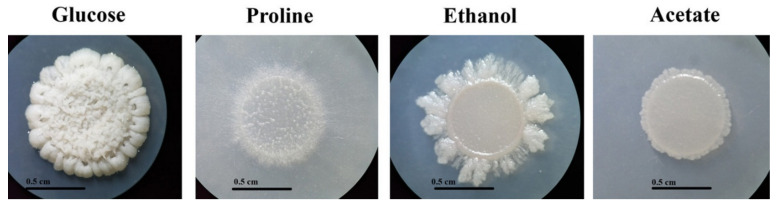
Yeast-like colony morphology of *T. marneffei* on gluconeogenic carbon sources. The 10^8^ conidia/5 μL of *T. marneffei* were spotted onto carbon-free medium containing 1.0% glucose, 50 mM proline, 0.5% ethanol, or 50 mM sodium acetate as the sole carbon source and incubated at 37 °C for 10 days. The yeast-like colony appearances on the gluconeogenic media were different from the yeast-like colony appearance on the glucose medium (Figure 2). The colony morphology revealed a white, raised yeast-like colony on the glucose medium. In contrast, the yeast-like colonies on the gluconeogenic carbon-containing media presented as flat and white with various shapes. The conidia could not germinate or elongate in the gluconeogenic carbon sources at 37 °C.

**Figure 3 jof-06-00102-f003:**
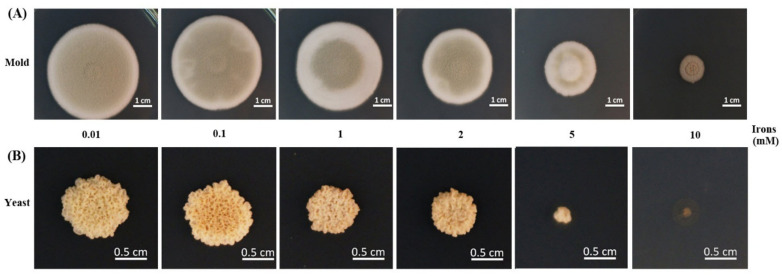
Growth of *T. marneffei* on solid media containing different iron concentrations. The 10^8^ conidia of *T. marneffei* were spotted onto glucose minimal medium containing various concentrations of ferric chloride (0.01–10.00 mM). The cultured were incubated at either 25 °C (**A**, mold) or 37 °C (**B**, yeast) for 12 days.

**Figure 4 jof-06-00102-f004:**
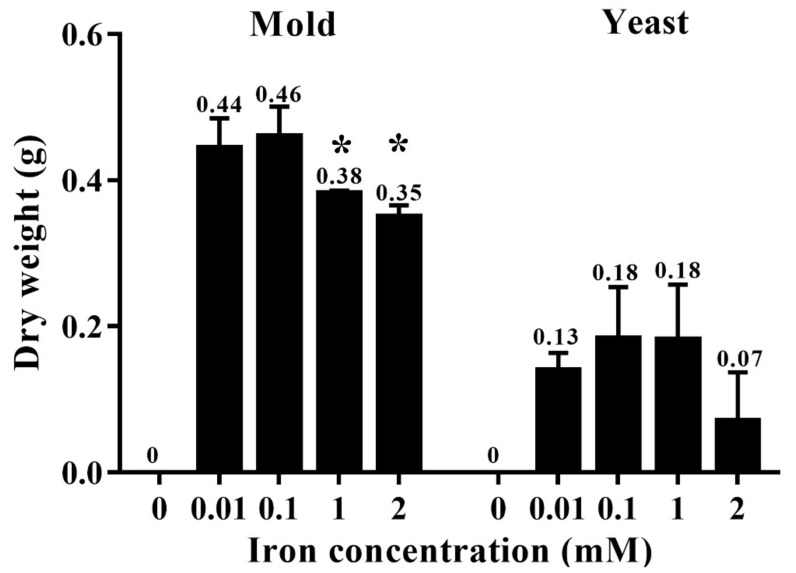
Growth of *T. marneffei* in liquid media containing different iron concentrations. The 10^8^ conidia/mL of *T. marneffei* was cultured in 100 mL minimal medium containing 0.00, 0.01, 0.10, 1.00, or 2.00 mM ferric chloride. The bars show average ± SD values from three independent experiments. Numbers above each bar indicate the average value. Asterisks show significant values at *p* < 0.05.

**Figure 5 jof-06-00102-f005:**
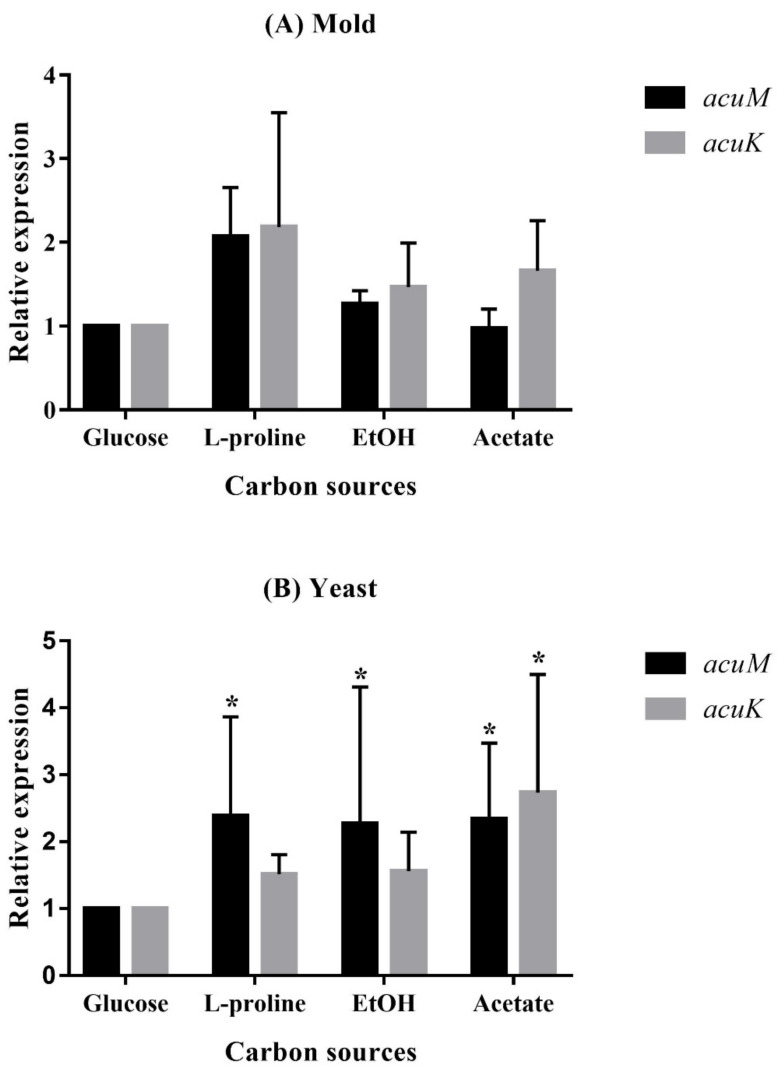
Relative expression levels of *acuM* and *acuK* during growth in gluconeogenic carbon sources. *T. marneffei* was cultured in the carbon-free medium containing one of the designated carbon sources, 1.0% glucose, 50 mM proline, 0.5% ethanol, or 50 mM acetate, for 36 h at 25 °C for the mold phase (**A**) and for 60 h at 37 °C for the yeast phase (**B**). The bars show average values and standard deviations from three independent experiments. An asterisk indicates a significant difference (*p* < 0.05) when compared to the respective gene in glucose conditions.

**Figure 6 jof-06-00102-f006:**
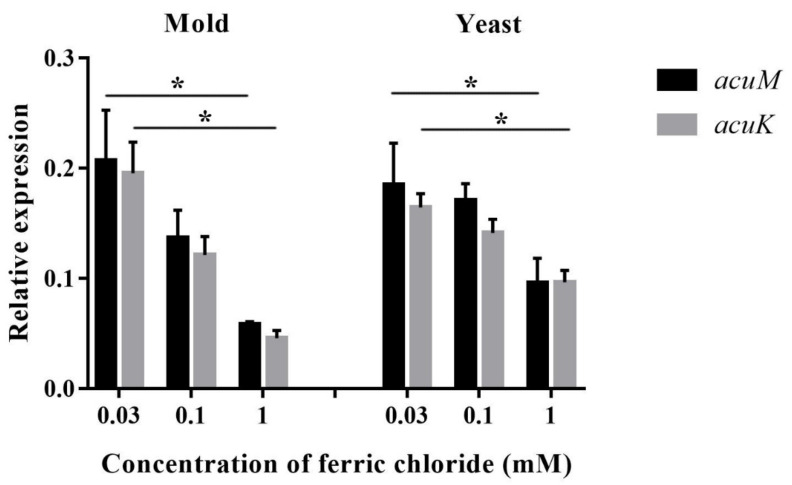
Relative expression levels of *acuM and acuK* during growth in various concentrations of iron. *T. marneffei* was cultured in the glucose minimal medium containing ferric chloride at different concentrations at 25 °C for 36 h (mold) or 37 °C for 60 h (yeast). An asterisk indicates a significant difference (*p* < 0.05) when compared to the respective gene at 0.03 mM iron concentration.

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
