# Peer review of "Expression of Talaromyces marneffei acuM and acuK Genes in Gluconeogenic Substrates and Various Iron Concentrations"

_jof, 2020, doi:10.3390/jof6030102_

Round 1

Reviewer 1 Report

Pongpom et al have examined the effects of carbon source (glucose vs. gluconeogenesis substrates) and iron concentration of mold and yeast growth forms of Talaromyces marneffeii. They correlate growth differences (Figures 1-4).  These data are clearly presented and represent interesting phenotypes. Both growth forms are severely reduced.

Under similar growth conditions varying as mentioned above, the expression of two transcription factors, AcuM and AcuK . These data are presented in Figures 5 & 6.

Figure 5 data on regulation of both transcription factors in various carbon sources to me does not demonstrate a clear relationship of acuK or acuM in mold growth. It seems that transcription of both genes is higher in gluconeogenic carbon sources versus glucose.  But the authors do not report p values for these data of (glucose vs all other substrates). This should be done.  

The authors state that increases in iron content cause a decrease in both transcription factors (Figure 6). The data show this clearly and point to regulation of growth by both transcription factors.

Perhaps the authors have this in mind, but knock-out phenotype mutants of both genes in conditions considered in this manuscript should be done.  A comparison of responses in these two genes should be compared to A. fumigatus.

In summary, the manuscript is clearly written and figures are easy to follow. 

Author Response

Thank you for you kind consideration. The manuscript has been revised according to two reviewers in the part of English language and introduction section improvement. Correction of grammatical errors was done by a certified personnel who is a native speaker working in research unit, Faculty of Medicine, Chiang Mai University.

Point by point response to reviewer 1’s comments is presented as follows:

Reviewer’s comment

Response to reviewer

Figure 5 data on regulation of both transcription factors in various carbon sources to me does not demonstrate a clear relationship of acuK or acuM in mold growth. It seems that transcription of both genes is higher in gluconeogenic carbon sources versus glucose.  But the authors do not report p values for these data of (glucose vs all other substrates). This should be done.   

We did the statistical analysis. However, unlike in the yeast phase, the transcript upregulation in the mold form was not met significant level (at the P < 0.05). This phenomenon can be explained by the different metabolisms in the different phases of this dimorphic fungud (Lie H, Zhang J, Li X et al. Identifying differentially expressed genes in the dimorphic fungus Penicillium marneffei by suppression subtractive hybridization. FEMS Microbiology Letters 2007: 270(1); 97-103). Additionally, our continuing study has found the pronounced effect after deletion of the acuK gene in the yeast than in the mold phase (this result will be reported after study completion), suggesting that the Acu transcription factor may play prominent role in the yeast phase and it needs further investigation.

Perhaps the authors have this in mind, but knock-out phenotype mutants of both genes in conditions considered in this manuscript should be done.  A comparison of responses in these two genes should be compared to A. fumigatus.

Thank you for the valuable suggestion. The phenotypic analysis on the deletion strains of both genes, including their target genes are currently under-investigating and it will be published as soon as the experiment done. Our initial result found that the function of these genes is very similar to those found in Aspergillus fumigatus, therefore this data will be compared with the previous results found in this fungus and other published liberated fungi.

Reviewer 2 Report

In this manuscript, the authors explore the role of transcription factors acuM and acuK in gluconeogenesis and iron homeostasis in the dimorphic fungi Talaromyces marneffei by evaluating fungal growth and acuM and acuK expression under gluconeogenic and increasing iron conditions. The manuscript does contain some grammar, spelling, and sentence structure errors.  The introduction, especially the first paragraph, is missing some references.  The introduction gives background on T. marneffei and indicates those who are at risk of infection.  It also provides sufficient background on the importance of gluconeogenesis and iron acquisition for fungal pathogens and sites that disruption of known gluconeogenesis regulators such as transcription factors acuM and acuK has been reported to attenuate virulence in other fungal pathogens.The authors also point out the reported involvement of acuM and acuK in iron acquisition and siderophore production in other fungal pathogens. The impact that this manuscript is likely to have is that it supports the conserved involvement of acuM and acuK in gluconeogenesis and iron metabolism in another genera of fungi.  The role of these two transcription factors were previously unknown in T. marneffei.  Functional follow-up experiments including mutation, intracellular survival, and virulence studies are needed to solidify the exact roles of these transcription factors as the authors point out in their discussion.

I have a few issues with this manuscript. I recommend having the manuscript edited to correct for spelling, grammar, and other errors (the “Results” section, for example, does not have a section 3.2).  This will allow for a smoother read.

Figure 4 legend does not state the number of replicates for each condition for the 3 independent experiments.  It also does not state what the bar graphs and error bars represent. Is it mean ± standard deviation?  Median± standard error?   Or something else? 

Figure 5 and 6 legends do not state the number of replicates and independent experiments performed nor does it state what the bar graphs and error bars represent, mean ± standard deviation or something else?

The Materials and Methods section states that the paired T-Test was used for statistical analysis.  Would the unpaired T-Test be more appropriate?

Section 3.3 could be expanded more to clearly state the observations.  The authors clearly state the observations for the sold medium but state “similar phenomenon happened in the dry mass determination assay.” Instead, state what was observed for the mold and yeast form (dry weight).  For example, state in the mold state, a statistically significant decrease in growth was observed at 1 and 2 mM iron.  In the yeast form, no significant decrease in growth was observed with increasing iron concentration up to 2 mM. 

In the discussion, the authors state that acuM and acuK from T. marneffei are more than 60% similar to the orthologs in A. nidulins and A. fumigatus at the gene level.  The authors state in the Materials and Methods section that similarity analysis was performed using NCBI Blast; however, no data is included in the manuscript results section.  The authors should state the IDs and sequences that were aligned and provide a figure of the alignment or BLAST results in the results or a supplementary data section.

Minor comments: after the first mention of Talaromyces marneffei, the authors should only write “T. marneffei”.

Author Response

Answer to reviewer 2

Reviewer’s comment

Response to reviewer

I recommend having the manuscript edited to correct for spelling, grammar, and other errors (the “Results” section, for example, does not have a section 3.2).  This will allow for a smoother read.

Correction of grammatical errors was done by a certified personnel who is a native speaker working in research unit, Faculty of Medicine, Chiang Mai University.

The mistake on missing section was corrected. The numbers of section were run smoothly in the revised manuscript.

Figure 4 legend does not state the number of replicates for each condition for the 3 independent experiments.  It also does not state what the bar graphs and error bars represent. Is it mean ± standard deviation?  Median± standard error?   Or something else? 

To explain about the Fig. 4 legend, we did culture only one tube to measure the dry weight, no replicate in each independent experiment. Therefore the bar showed the average value and standard deviation from 3 independent experiments. The statement to indicate what the bar graphs and error bars represent is revised to “The bars showed average±SD values from three independent experiments”

Additionally, the confusion on the numbers above each bar was corrected by adding the sentence “Numbers above each bar indicated the average value”. We assured that this should be enough for clarification.

Figure 5 and 6 legends do not state the number of replicates and independent experiments performed nor does it state what the bar graphs and error bars represent, mean ± standard deviation or something else?

The explanation for this question is same to the Fig.4 legend. The Fig. 5 and 6 legends were then clarified with stating the bar indication of mean ± standard deviation.

The Materials and Methods section states that the paired T-Test was used for statistical analysis.  Would the unpaired T-Test be more appropriate?

We did compare each tested condition with the control (glucose in gluconeogenetic-relating condition, and with 0.03 mM iron in the iron condition) in the same fungus (wild type). After consultation with our expert, we selected the paired T-test since it is used to compare 2 conditions in one genetic background. We therefore confident that the paired T-test is appropriated in this study.

Section 3.3 could be expanded more to clearly state the observations.  The authors clearly state the observations for the sold medium but state “similar phenomenon happened in the dry mass determination assay.” Instead, state what was observed for the mold and yeast form (dry weight).  For example, state in the mold state, a statistically significant decrease in growth was observed at 1 and 2 mM iron.  In the yeast form, no significant decrease in growth was observed with increasing iron concentration up to 2 mM. 

The explanation of the dry mass determination assay was revised into “Likewise, similar results were observed in the dry mass determination assay. In the mold phase, a significant decreased in growth was observed at 1 and 2 mM iron. In the yeast form the reduction in cell mass with increasing iron concentration up to 2 mM was observed even though without significant (Fig.4)” (page 5, line 17-20 in the revised manuscript)

In the discussion, the authors state that acuM and acuK from T. marneffei are more than 60% similar to the orthologs in A. nidulins and A. fumigatus at the gene level.  The authors state in the Materials and Methods section that similarity analysis was performed using NCBI Blast; however, no data is included in the manuscript results section.  The authors should state the IDs and sequences that were aligned and provide a figure of the alignment or BLAST results in the results or a supplementary data section.

We decided to add the gene analysis and alignment result in the supplementary data section since it is not the main result for this study.

Minor comments: after the first mention of Talaromyces marneffei, the authors should only write “T. marneffei”.

We revised the manuscript for this suggestion. The “T. marneffei” was used instead of Talaromyces marneffei wherever it appeared in the appropriate position i.e. page 2, Line 8; page 3, Line 1; page 3, Line 40; page 3, Line 40; page 4, Line 11; page 5, Line 25; page 6, Line 2; page 9, Line 7.